# Mix Early, Forget Less: Data Mixing During Pretraining Builds Resistance to Forgetting

**Lawrence Feng,**[*] **Gaurav Rohit Ghosal, Jacob Mitchell Springer,**
**Ziqian Zhong**, **Aditi Raghunathan**
Carnegie Mellon University

## Abstract

After web-scale pretraining, language models are often further trained to add domain skills and behaviors, and later fine-tuned to ingest new data or meet specific downstream requirements. A persistent challenge in such sequential pipelines is catastrophic forgetting: later training can degrade previously learned capabilities. Prior mitigation strategies largely focus on fine-tuning-time interventions and treat the upstream training procedure as fixed. We show that upstream data placement matters: mixing a small amount (a few % of the overall pretraining mixture) of capability-relevant data into pretraining builds resistance to forgetting, yielding substantially better learning–retention tradeoffs under subsequent training than introducing the domain only after pretraining. We demonstrate this effect across multiple settings, including specialized domain adaptation and instruction tuning. We also study algorithmic choices during continual pretraining and find that dropout and data replay provide additional gains that are consistently complementary to pretraining-time mixing.

## 1 Introduction

In this work, we study a canonical three-stage lifecycle: a base model is first pretrained on general web data, then specialized to a target domain via continual pre-training (CPT), and finally subjected to downstream fine-tuning (FT) on a different task. Our focus is the learning–retention tradeoff: how much specialized capability is retained after later training, at a given level of performance on the downstream task. While failures in retention are well-documented, most mitigation strategies aim to reduce interference during fine-tuning—such as replay (Bethune et al., 2025), elastic weight consolidation (Jhajj & Lin, 2025), or parameter-efficient isolation (Lin et al., 2025)—often treating the upstream pretraining and specialization procedure as fixed. In doing so, they primarily target fine-tuning dynamics rather than the upstream training choices that determine how specialized knowledge is represented.

Mixing a small fraction of domain data into pretraining consistently improves the attainable learning–retention tradeoff relative to end-loaded specialization. This challenges the intuition that models forget early training data (recency bias) and should therefore encounter specialized knowledge only at the end (Wei et al., 2025). Moreover, the advantage of mixing can be latent: it may not improve immediate post-CPT performance, yet it yields substantially better retention after subsequent training. In Section 3.2, we examine how much of the specialized domain to mix into pretraining and if mixing remains optimal under a fixed compute budget.

We then turn our attention to studying the impact of *algorithmic choices during CPT* on the learning–retention tradeoff. We look at techniques such as dropout, pretraining data replay, and LoRA (Hu et al., 2021).

## 2 Preliminaries and Setting

Our focus is to train models that are resistant to forgetting under *subsequent adaptation*. Concretely, a model developer may (1) pretrain a base model on a broad corpus, (2) adapt it to a specialized

---

[*]Corresponding author: `lawrencefeng@cmu.edu`

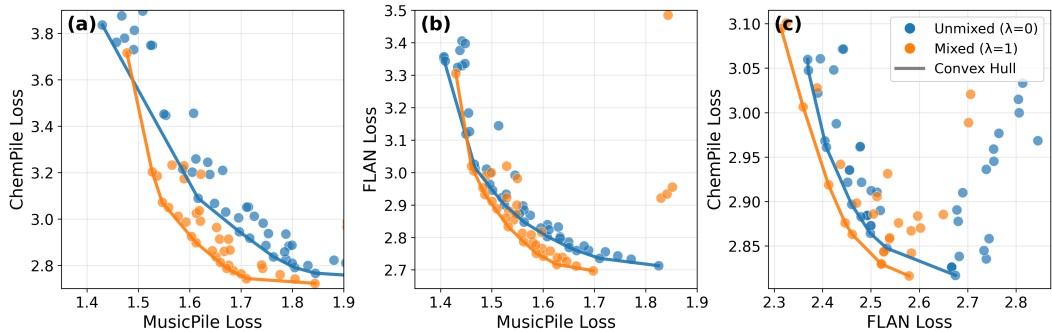

Figure 1: **Data mixing improves long-term retention across post-training pipelines.** Each point is a separate Stage-2 CPT run (different hyperparameters); solid lines trace the Pareto frontier for each initialization. **(a,b)** Starting from either an *unmixed* base model ($\lambda=0$, pretrained on C4) or a *mixed* base model ($\lambda=1$, pretrained on C4 with MusicPile mixed in), we perform CPT on MusicPile and then apply Stage-3 fine-tuning on **ChemPile** (a) or **FLAN** (b), reusing the *same* CPT checkpoints across panels. **(c)** A separate pipeline where the specialized domain is **FLAN**: we pretrain on C4 with/without FLAN mixing, CPT on FLAN, then fine-tune on **ChemPile**.

domain, and then (3) ship the model to downstream users who further fine-tune it for their own purposes. We model these stages with three datasets: $\mathcal{D}_{\text{gen}}$ (general pretraining), $\mathcal{D}_{\text{spec}}$ (specialized domain), and $\mathcal{D}_{\text{ft}}$ (downstream adaptation).

**Stage 1: Mixed pretraining.** In mixed pretraining, the model is trained on a mixture of a general corpus $\mathcal{D}_{\text{gen}}$ and a fixed amount of specialized data from $\mathcal{D}_{\text{spec}}$. We refer to the resulting mixed pretraining corpus as

$$\mathcal{D}_{\text{mix}}(\lambda) \;=\; \mathcal{D}_{\text{gen}} \;+\; (\lambda \mathcal{D}_{\text{spec}}),$$

where $\lambda \in [0,1]$ controls how much specialized data is included. We denote the parameters after mixed pretraining by $\theta_{\text{pt}}$.

**Stage 2: Continual Pretraining (CPT).** Starting from the pretrained parameters $\theta_{\text{pt}}$, we continue training exclusively on the specialized corpus $\mathcal{D}_{\text{spec}}$; this adaptation can induce regression on previously learned general capabilities. In the default setting, we apply early stopping and train on $\mathcal{D}_{\text{spec}}$ until the validation loss stops improving (with a maximum budget of 2B tokens), yielding We measure specialized performance immediately afterward by the *immediate loss* $\mathcal{L}_{\text{im}} := \mathcal{L}(\theta_{\text{cpt}}; \mathcal{D}_{\text{spec}})$.

**Stage 3: Downstream Fine-tuning.** In practice, models are rarely "done" after CPT: downstream users often further adapt them to new objectives, such as instruction tuning or continued domain adaptation on a user-specific corpus. We model this as downstream fine-tuning on a dataset $\mathcal{D}_{\text{ft}}$, and define the downstream loss $\mathcal{L}_{\text{ft}} := \mathcal{L}(\theta_{\text{ft}}; \mathcal{D}_{\text{ft}})$.

Further training can degrade previously acquired capabilities, so the specialized loss on $\mathcal{D}_{\text{spec}}$ may increase relative to $\mathcal{L}_{\text{im}}$. We therefore measure specialized retention after downstream adaptation by the *retained loss* $\mathcal{L}_{\text{ret}} := \mathcal{L}(\theta_{\text{ft}}; \mathcal{D}_{\text{spec}})$. Our main goal is to understand how choices in how $\mathcal{D}_{\text{spec}}$ is incorporated in Stages 1–2 affect $\mathcal{L}_{\text{ret}}$ under subsequent downstream fine-tuning.

## 2.1 Evaluation Methodology

**Loss frontiers.** Sweeping CPT methods and hyperparameters yields a set of runs with different tradeoffs between downstream performance and specialized retention, summarized by points $(\mathcal{L}_{\text{ft}}, \mathcal{L}_{\text{ret}})$. We summarize the best attainable tradeoffs using the *loss frontier*: the subset of non-dominated runs for which no other run achieves lower loss on both axes. We say one method (or base model) *dominates* another if its frontier achieves equal or lower $\mathcal{L}_{\text{ret}}$ at matched $\mathcal{L}_{\text{ft}}$ over the range covered by our sweeps.

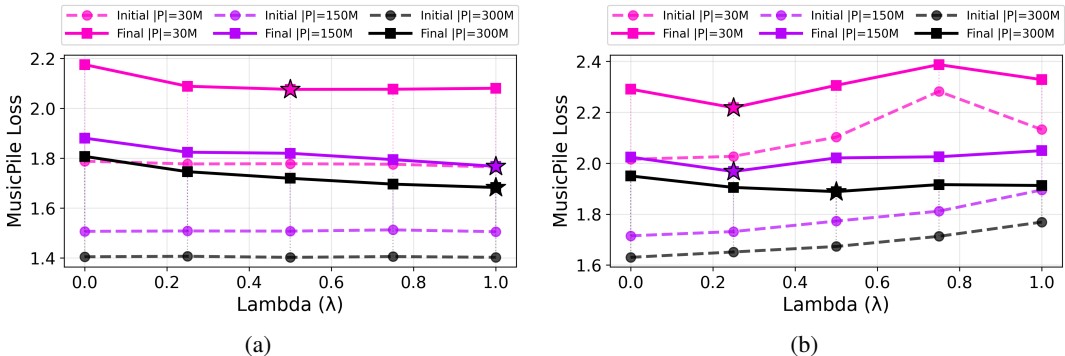

(a)  (b)

Figure 2: **Mixing remains optimal for retention under a fixed MusicPile compute budget.** We pretrain models with a $\lambda$ fraction of MusicPile mixed into C4, while holding total MusicPile exposure fixed. Dashed curves show *immediate* MusicPile loss after CPT; solid curves show *retained* MusicPile loss after subsequent fine-tuning. Across budgets, increasing $\lambda$ worsens immediate post-CPT loss, but improves retained loss, indicating that gains from end-loaded CPT are brittle.

## 3 EXPERIMENTS AND RESULTS

### 3.1 MIXED PRETRAINING MITIGATES FORGETTING

**Main result.** Figure 1 shows the resulting loss frontiers. In both downstream regimes, mixed pretraining improves the attainable tradeoff: at matched downstream loss, models derived from the mixed base achieve lower retained loss on MusicPile than those derived from the unmixed base. Mixing does not eliminate the learning–retention tradeoff, but shifts the frontier outward, enlarging the set of achievable operating points. We additionally find that the benefits of mixing hold for preserving instruction tuning (Figure 1 (c)) when considering $\mathcal{D}_{\text{spec}} = \text{FLAN}$.

### 3.2 FINE-GRAINED INVESTIGATION OF MIXING IN PRETRAINING

Having established that mixed pretraining shifts the downstream–retention frontier, we next ask: (i) how much early exposure to $\mathcal{D}_{\text{spec}}$ is needed, and (ii) does early mixing help when the total $\mathcal{D}_{\text{spec}}$ compute budget is held fixed?

#### 3.2.1 HOW MUCH $\mathcal{D}_{\text{SPEC}}$ SHOULD WE MIX?

**Impact of Mixture Fraction** In Figure 2(a), we show the retained and immediate MusicPile losses as a function of the pretraining mixture fraction $\lambda$. We observe retained MusicPile loss $\lambda$ decreases as $\lambda$ increases, suggesting that increasing the ratio of specialized data during pretraining is generally helpful.

**Benefits of Mixing Can Be Latent.** In Figure 2(a), we additionally plot the *immediate* MusicPile loss $\mathcal{L}_{\text{im}}$. Strikingly, the choice of mixture fraction $\lambda$ has little effect on $\mathcal{L}_{\text{im}}$. Thus, CPT can drive models to comparable specialized performance immediately after CPT even when they differ in how much $\mathcal{D}_{\text{spec}}$ was seen during initial pretraining. However, these checkpoints behave very differently under subsequent adaptation, as reflected by the spread in retained loss $\mathcal{L}_{\text{ret}}$.

#### 3.2.2 IS EARLY MIXING STILL BENEFICIAL UNDER A FIXED $\mathcal{D}_{\text{SPEC}}$ COMPUTE BUDGET?

We now study a compute-matched variant that fixes the total number of $\mathcal{D}_{\text{spec}}$ tokens seen across pretraining and CPT and varies only their allocation between the stages.

**Mixing worsens immediate performance in the compute-matched setting.** Figure 2(b) shows that the immediate specialized loss $\mathcal{L}_{\text{im}}$ increases with $\lambda$ in the compute-matched setting. When the total $\mathcal{D}_{\text{spec}}$ budget is fixed, allocating more of $\mathcal{D}_{\text{spec}}$ to Stage 1 mixing yields worse post-CPT MusicPile performance.

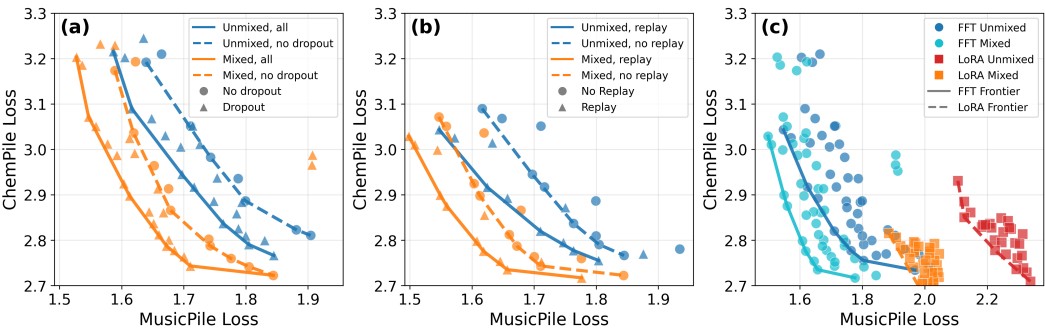

Figure 3: **CPT-time interventions (MusicPile → ChemPile).** We compare loss frontiers between downstream ChemPile loss $\mathcal{L}_{\mathrm{ft}}$ (x-axis) and retained MusicPile loss $\mathcal{L}_{\mathrm{ret}}$ (y-axis) for mixed vs. unmixed Stage 1 checkpoints under different Stage 2 CPT procedures. **(Left)** Dropout during CPT ($p \in \{0, 0.02, 0.05\}$) improves the frontier for both base models. **(Middle)** Replay during CPT (0 vs. 1% $\mathcal{D}_{\mathrm{gen}}$ tokens interleaved) improves retention at matched downstream loss. **(Right)** LoRA-based CPT shows a markedly larger mixed–unmixed gap than full-parameter CPT, though the resulting LoRA frontiers are overall dominated by full-parameter CPT.

**Mixing improves retention under subsequent adaptation.** Despite harming $\mathcal{L}_{\mathrm{im}}$, early mixing improves retention after downstream training. As $\lambda$ increases, the retained loss $\mathcal{L}_{\mathrm{ret}}$ decreases and the degradation gap $\mathcal{L}_{\mathrm{ret}} - \mathcal{L}_{\mathrm{im}}$ shrinks, indicating that specialized gains from CPT are more brittle under further training than gains supported by early exposure.

## 3.3 STUDYING THE IMPACT OF CONTINUAL PRETRAINING METHODOLOGY

### 3.3.1 DROPOUT AND PRETRAINING MIXING ARE COMPLEMENTARY

Dropout is a standard regularizer, but it is often disabled in modern decoder-only LM training, making it a relatively underexplored knob for continual pretraining. We hypothesize that introducing small amounts of dropout during CPT encourages more robust representations of $\mathcal{D}_{\mathrm{spec}}$, improving retention under subsequent adaptation.

**Results.** Dropout improves the downstream–retention tradeoff for both mixed and unmixed base models, shifting their frontiers outward (Figure 3, left); however, it does not fully close the gap between them. This suggests that CPT-time regularization and pretraining-time mixing provide complementary benefits: dropout improves robustness of $\mathcal{D}_{\mathrm{spec}}$ acquisition within Stage 2, while mixing changes the base model in a way that remains advantageous even with the same CPT intervention.

### 3.3.2 REPLAY DATA AND PRETRAINING MIXING ARE COMPLEMENTARY

**Results.** Replay improves the downstream–retention tradeoff for both mixed and unmixed base models (Figure 3, middle). At matched downstream loss $\mathcal{L}_{\mathrm{ft}}$, replay yields lower retained loss $\mathcal{L}_{\mathrm{ret}}$, indicating improved resistance to forgetting under subsequent adaptation. However, replay does not substitute for pretraining-time mixing: over most of the frontier, unmixed models with replay are still dominated by mixed models without replay. Combining mixing and replay yields the best tradeoffs, suggesting these interventions are complementary.

## 4 DISCUSSION

Overall, our results show that upstream decisions can substantially influence the attainable learning–retention tradeoff under subsequent training. In particular, mixing specialized data into pretraining dramatically increases robustness to future forgetting. While regularization and replay during continual pretraining can partially improve retention, they do not eliminate the fragility of capabilities introduced only after pretraining. These findings offer concrete guidance for how model developers can preemptively mitigate catastrophic forgetting—an axis that remains relatively underexplored compared to fine-tuning-time interventions.

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

## A    RELATED WORKS

**Catastrophic Forgetting** A recurring challenge in sequential training is *catastrophic forgetting*: when a model is optimized on new data, its performance can deteriorate on behaviors it previously exhibited McCloskey & Cohen (1989). For language models, this phenomenon shows up in modern training pipelines. For example, instruction tuning and RLHF can trade off against preexisting capabilities, an effect often discussed as an "alignment tax" Ouyang et al. (2022). Relatedly, several works show that behaviors introduced during safety finetuning can be quickly weakened or reversed by subsequent training on different objectives or data Yang et al. (2023); Qi et al. (2023). Forgetting-like tradeoffs also appear in adjacent settings such as knowledge editing Nishi et al. (2025) and unlearning Maini et al. (2024a). Beyond documenting the effect, recent work has started to map how training choices shape its severity: for instance, LoRA-style adaptation can alter forgetting dynamics Biderman et al. (2024), and longer pretraining can change how brittle or persistent acquired capabilities are Springer et al. (2025). In this paper, we focus on catastrophic forgetting in *sequential domain adaptation*, and ask a finer-grained question: what properties of a domain-adapted checkpoint determine whether domain knowledge persists under subsequent training?

**Training Dynamics of LLMs** A significant amount of prior works aim to characterize the learning dynamics of pretraining. Leybzon & Kervadec (2024) find that pretraining-time memorization undergoes cycles of learning and forgetting. Similarly, Wei et al. (2025) that memorized content seen earlier in pretraining can be diluted throughout training, while content seen later in pretraining remains more easily accessible. Other works have studied the dynamics of multiple stages of training. Qi et al. (2025); Liu et al. (2025) study the role of mid-training in LLM pipelines, showing that it helps bridge the distributional differences between pre and post-training. In this work, we study the problem of retaining specialized domain knowledge and demonstrate that mixing data in pre-training can have important benefits.

**Pretraining Interventions** Prior works examine pre-training time interventions for enforicng desired downstream model properties. Maini et al. (2025); O'Brien et al. (2025) propose filtering and augmenting data during pre-training to improve safety. Similarly, Sam et al. (2026) demonstrate that the impact of such interventions improves as they are introduced earilier in pre-training. Beyond safety, Maini et al. (2024b) shows that rephrasing web data can improve loss and zero-shot capabilities. While these works incorporate downstream tasks during pre-training, they extensively modify the pre-training corpus by incorporating data-augmentations and filtering of the dataset. We show that *merely* mixing domain-specific data during the initial pre-training phase can mitigate catastrophic forgetting.

# B  Concrete Experimental Details

**Model and token budgets.** Across all experiments, we use SmolLM2-135M and perform Stage 1 pretraining on a fixed 10B-token stream from $\mathcal{D}_{\text{gen}}$, together with an additional $\lambda$-fraction of the available specialized corpus $\mathcal{D}_{\text{spec}}$. In Stage 2, we apply early stopping on $\mathcal{D}_{\text{spec}}$ with a maximum budget of 2B tokens, except in compute-matched ablations where the $\mathcal{D}_{\text{spec}}$ budget is fixed by construction. In Stage 3, we fine-tune on $\mathcal{D}_{\text{ft}}$ for a fixed token budget specific to each setting (200M tokens in our main ChemPile fine-tuning runs).

**Dataset instantiations.** We study two instantiations of the three-stage pipeline. In all cases, the general pretraining corpus is $\mathcal{D}_{\text{gen}}$ (10B tokens subsampled from C4), and we vary how $(\mathcal{D}_{\text{spec}}, \mathcal{D}_{\text{ft}})$ are instantiated.

**Optimization and hyperparameters.** Unless otherwise stated, we train with AdamW using linear warmup followed by cosine decay and fix weight decay to the LitGPT default AI (2023) of 0.1. Our base CPT sweep varies learning rate (5 values), batch size (3 values), and dropout (3 values)[1].

**Specialized-domain setting.** We set $\mathcal{D}_{\text{spec}}$ to MusicPile and evaluate retention under two downstream adaptation regimes by setting $\mathcal{D}_{\text{ft}}$ to either more C4, FLAN (instruction tuning), or ChemPile (continued domain training). For MusicPile, we consider varying-sized subsets to study data scarcity (30M–300M tokens).

**Instruction-tuning setting.** We set $\mathcal{D}_{\text{spec}}$ to FLAN (Stage 2 instruction tuning) and set $\mathcal{D}_{\text{ft}}$ to ChemPile (Stage 3 domain fine-tuning). This setting models benign downstream fine-tuning for domain capability that can nevertheless erode instruction-following or behavioral tuning.

## B.1  Experimental Setups in Section 3

### B.1.1  Setup in Section 3.1

**Setup.** We compare mixed and unmixed pretraining for the same specialized corpus $\mathcal{D}_{\text{spec}} = $ MusicPile, and evaluate retention under two downstream adaptation regimes: $\mathcal{D}_{\text{ft}} = $ ChemPile and $\mathcal{D}_{\text{ft}} = $ FLAN, as well as a setting where $\mathcal{D}_{\text{spec}} = $ FLAN and $\mathcal{D}_{\text{ft}} = $ ChemPile.

### B.1.2  Setup in 3.2.1

**Experimental setup.** We fix the Stage 2 CPT procedure and vary only the Stage 1 allocation of MusicPile into pretraining. Concretely, we pretrain on $\mathcal{D}_{\text{mix}}(\lambda) = \mathcal{D}_{\text{gen}} + \lambda\mathcal{D}_{\text{spec}}$ with $\mathcal{D}_{\text{spec}} = $ MusicPile and $\lambda \in \{0.0, 0.25, 0.5, 0.75, 1.0\}$, then run CPT on MusicPile till saturation to obtain $\theta_{\text{cpt}}$. To measure retention under subsequent adaptation, we set $\mathcal{D}_{\text{ft}} = \mathcal{D}_{\text{gen}}$ (continued pretraining on C4) and report both $\mathcal{L}_{\text{im}}$ and $\mathcal{L}_{\text{ret}}$ on MusicPile.

### B.1.3  Setup in 3.2.2

**Experimental setup.** We pretrain on $\mathcal{D}_{\text{mix}}(\lambda) = \mathcal{D}_{\text{gen}} + \lambda\mathcal{D}_{\text{spec}}$ and then run CPT on the remaining $(1 - \lambda)\mathcal{D}_{\text{spec}}$, so that every model sees exactly one pass over $\mathcal{D}_{\text{spec}}$ across Stages 1–2. Thus, $\lambda$ controls how a fixed $\mathcal{D}_{\text{spec}}$ budget is allocated between early mixing (Stage 1) and dedicated CPT (Stage 2), with $\lambda = 1$ corresponding to mixing-only (no CPT).

### B.1.4  Setup in 3.3.1

**Setup.** We study dropout as a CPT-time intervention by enabling dropout during Stage 2 and varying the dropout rate $p \in \{0.0, 0.02, 0.05\}$. Initial experiments indicated any larger $p$ led to worse post-CPT loss. For each $p$, we sweep the base CPT optimization grid described in Section 2.1 (learning rate and batch size, with weight decay and warmup fixed), starting from both mixed and unmixed Stage 1 checkpoints, and compare the resulting loss frontiers (Figure 3 and **??**, left).

---

[1]Learning rates: $\{10^{-4}, 2\times10^{-4}, 5\times10^{-4}, 10^{-3}, 5\times10^{-3}\}$; batch sizes: $\{192, 480, 896\}$; dropout: $\{0, 0.02, 0.05\}$. Warmup steps fixed to 100.

### B.1.5   SETUP IN 3.3.2

**Setup.** We study rehearsal during CPT by injecting a small amount of pretraining data into Stage 2, following Bethune et al. (2025). In this ablation (run for the 300M-token MusicPile setting), during CPT on $\mathcal{D}_{\text{spec}}$ we either use no replay or interleave an additional 1% of $\mathcal{D}_{\text{gen}}$ tokens relative to the $\mathcal{D}_{\text{spec}}$ CPT budget. We sweep the same base CPT optimization grid (Section 2.1) from both mixed and unmixed Stage 1 checkpoints and compare loss frontiers (Figure 3, middle).

## C   EXTENDED RESULTS

### C.0.1   LORA PRODUCES A WORSE FRONTIER, WHILE AMPLIFYING MIXING EFFECTS

**Setup.** We study parameter-efficient CPT by replacing full-parameter updates in Stage 2 with LoRA updates. We fix the LoRA configuration (rank $r=64$, scaling $\alpha=128$) and sweep the base CPT optimization grid (Section B) starting from both mixed and unmixed Stage 1 checkpoints, comparing the resulting loss frontiers to full-parameter CPT (Figure 3, right).

**Result.** LoRA traces a qualitatively different downstream–retention tradeoff from full-parameter CPT. Relative to full-parameter CPT, LoRA tends to achieve lower downstream ChemPile loss $\mathcal{L}_{\text{ft}}$ but worse retained MusicPile loss $\mathcal{L}_{\text{ret}}$. Consequently, the LoRA frontiers are overall dominated by the full-parameter (FFT) frontiers for both mixed and unmixed base models. At the same time, the effect of pretraining-time mixing is amplified under LoRA: the separation between mixed and unmixed LoRA frontiers is substantially larger than in the full-parameter setting, suggesting that early exposure to $\mathcal{D}_{\text{spec}}$ becomes more important when Stage 2 adaptation is constrained to a low-rank subspace. Interestingly, we also find that the change in loss on $\mathcal{D}_{\text{spec}}$ before and after finetuning is much greater for LoRA compared to FFT (see Figure 4).

## D   ADDITIONAL PLOTS

### D.1   LORA VS FFT DURING CPT

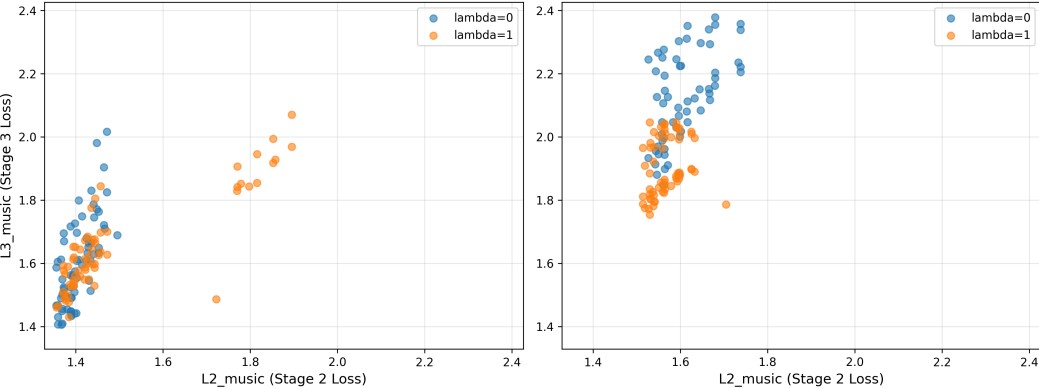

Figure 4: Stage 3 MusicPile loss vs Stage 2 Musicpile loss. We see a greater downward shift between blue and orange points in the LoRA plotting, suggesting a greater resistance to forgetting from mixing. Notably, the point clouds are mostly vertically stacked, indicating similar stage 2 performance across hyperparameters.

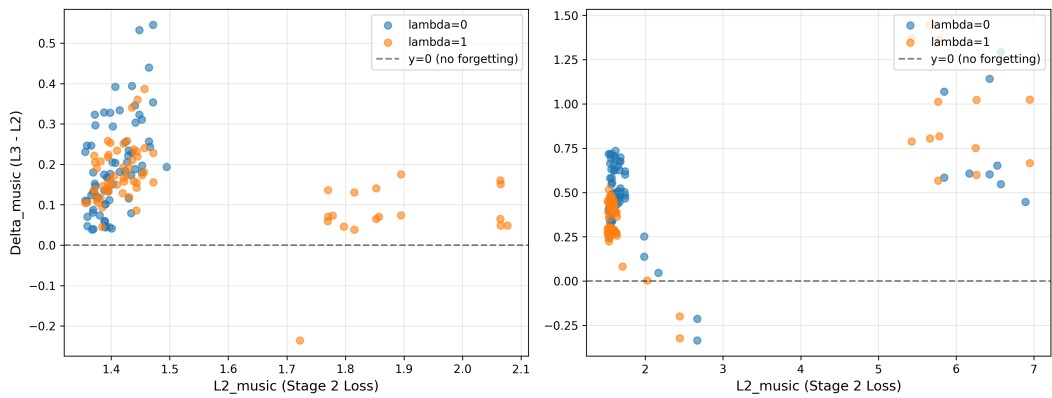

Figure 5: Similar to the figure above, but with delta loss on the y-axis

## D.2  REPLAY AND DROPOUT

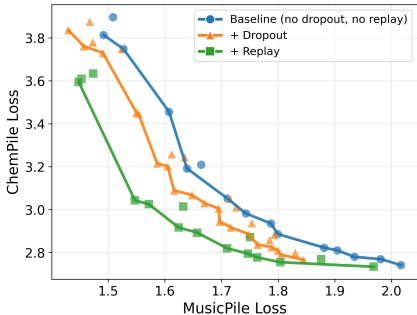

Figure 6: Comparison in frontier shift from baseline when using dropout vs replay.

