# OpenReview forum: "Mix Early, Forget Less: Data Mixing During Pretraining Builds Resistance to Forgetting"
_ICLR.cc/2026/Workshop/GRaM — ICLR 2026 Workshop GRaM Poster_

### Official Review · Reviewer_rHVD · 2026-02-15

**Rating:** 6
**Confidence:** 3

**Review:**

This paper presents a clear and empirically supported study showing that early mixing of domain-specific data into pretraining significantly improves retention under subsequent downstream fine-tuning. While it does not explicitly use geometric modeling, the results are relevant to loss landscape geometry and challenging assumptions about training dynamics. The contribution is novel through its empirical methodology and ablations, though the approach is conceptually simple and primarily empirical rather than formal or theoretical.

  The paper addresses an implicit geometric phenomenon by showing how early mixing of domain data into pretraining changes the structure of the representation space, resulting in improved robustness under downstream fine-tuning.

  Strengths
-The core proponent of this research, mixing a small fraction of domain data into pretraining rather than introducing it only during continual pretraining, is conceptually simple yet demonstrates a significant effect on retention under downstream fine-tuning.

   -Experiments are well-designed and include multiple datasets, downstream adaptations, and continual pretraining interventions, supporting the robustness of the conclusions.

   -The paper is clearly written, well-organized, and figures effectively illustrate the main findings.

Weakness
  -The paper does not directly incorporate explicit geometric methods (e.g., non-Euclidean metrics, equivariant operators, or manifold-preserving embeddings).

  -All experiments use a small-scale model (SmolLM2-135M), which limits generalizability to larger models.

   -The connection to geometry is only implicit, and the discussion could be improved by explicitly framing the results in terms of representation geometry or loss landscape effects.

**Pmlr Suitability:**

NA

---

### Official Review · Reviewer_fp5u · 2026-02-19
**Early Data Mixing Improves Retention Under Sequential Adaptation**

**Rating:** 7
**Confidence:** 3

**Review:**

This paper presents a clean empirical study demonstrating that mixing a small fraction of domain-specific data into pretraining significantly improves retention under subsequent downstream fine-tuning. The work is well-designed and empirically convincing, though limited in scale and lacking a mechanistic explanation for the observed effect.

This paper studies catastrophic forgetting in a three-stage language model pipeline:

1. Pretraining on general data D_gen
2. Continual pretraining on a specialized domain D_spec
3. Downstream fine-tuning on D_ft

The central claim is that incorporating a small fraction of D_spec during Stage 1 pretraining — rather than introducing it only during Stage 2 — significantly improves long-term retention after Stage 3 adaptation.

Retention is evaluated using the downstream–retention Pareto frontier over pairs:

    (L_ft, L_ret)

The authors show that early mixing shifts this frontier outward across multiple dataset configurations. Importantly, the gains from mixing can be latent: immediate post-CPT loss L_im may not improve, yet retained loss L_ret after Stage 3 fine-tuning is substantially lower.

A compute-matched ablation demonstrates that retention gains persist even when total exposure to D_spec is held fixed, indicating that the effect is not merely due to increased domain token count. Additional experiments show that dropout and replay during continual pretraining are complementary to pretraining-time mixing.

Quality

The empirical methodology is strong. The frontier-based evaluation avoids cherry-picking and provides a principled comparison of learning–retention tradeoffs. The compute-matched ablation is particularly convincing.

However, all experiments are conducted at 135M scale. It remains unclear whether the effect persists at larger model sizes. Frontier dominance claims are primarily visual and would benefit from uncertainty quantification.

Clarity

The paper is clearly written and logically structured. The three-stage pipeline is easy to follow and figures effectively illustrate frontier shifts.

Originality

The intervention is conceptually simple, but framing it in terms of long-term retention under sequential adaptation is novel. The compute-controlled allocation experiment strengthens the originality.

The work does not introduce new geometric tools, though it is relevant to representation geometry and loss landscape structure.

Significance

The findings are practically relevant for model developers managing multi-stage training pipelines. Upstream data allocation is shown to materially affect downstream robustness to forgetting.

Broader impact would benefit from:
- Larger-scale validation
- Mechanistic or geometric analysis
- Statistical robustness testing

Strengths

1. Clear and focused research question.
2. Strong compute-controlled ablation.
3. Robust hyperparameter sweeps.
4. Demonstration of complementarity with dropout and replay.

Weaknesses

1. Limited to small-scale (135M) models.
2. Lacks mechanistic explanation.
3. No statistical uncertainty estimates on frontiers.
4. Limited negative controls (e.g., unrelated-domain mixing).

Overall:
The paper provides a clean and useful empirical result, though theoretical depth and scaling validation would strengthen it.

**Pmlr Suitability:**

NA

---

### Official Review · Reviewer_6GXP · 2026-02-22
**Mix Early, Forget Less: Data Mixing During Pretraining Builds Resistance to Forgetting**

**Rating:** 5
**Confidence:** 3

**Review:**

This paper examines catastrophic forgetting in a three-stage sequential training pipeline: (1) pretraining on general web data, (2) continual pretraining on target-domain data, and (3) fine-tuning on downstream tasks. The authors propose mitigating forgetting by mixing a portion of domain-specific data into the pretraining stage, rather than relying primarily on fine-tuning-based mitigation strategies. The goal is to preserve specialized capabilities while adapting to the target domain.

#### Strengths

1. The core idea is simple, intuitive, and clearly explained.
2. The paper provides empirical results showing improved training loss compared to a baseline that performs pretraining without data mixing.
3. The method yields incremental gains on top of existing techniques such as replay and dropout, suggesting the improvement is not merely duplicating what those baselines already achieve.

#### Weaknesses

1. The ablation over different data distributions is limited, which makes it difficult to assess how broadly the proposed approach generalizes and under what conditions it is most beneficial.
2. The paper would have been stronger with more explicit geometric or representation-level insights (e.g., how mixing changes feature drift, gradient alignment, or representation collapse), which would better align with the workshop’s theme.
3. The experiments are limited to relatively small models (e.g., 135M parameters), so it’s unclear whether the benefits of data mixing will persist at larger scales.

**Pmlr Suitability:**

NA

---

### Meta-Review · Area_Chair_fTaq · 2026-02-23

**Decision:**

Accept

**Metareview:**

The reviews are pretty positive and all agree on both strengths and weaknesses. All concur that the paper is well-written with well designed experiments. One weakness pointed out is that the models are relatively small, which I think is fine for a workshop paper.
The second weakness is that the geometric connection could be made explicit which would make it a better fit for the workshop. However, reviewers also agree that it is relevant to representation geometry and loss landscapes. For this reason, I recommend accepting the paper.

**Relevance To Proceedings:**

Tiny paper — does not apply

**Relevance To Workshop:**

Yes — suitable for GRaM

---

### Decision · Program_Chairs · 2026-03-02

Accept (Poster)